# Parental Disease Specific Knowledge and Its Impact on Health-Related Quality of Life

**DOI:** 10.3390/children9010098

**Published:** 2022-01-11

**Authors:** Luisa Stasch, Johanna Ohlendorf, Ulrich Baumann, Gundula Ernst, Karin Lange, Christiane Konietzny, Eva-Doreen Pfister, Kirsten Sautmann, Imeke Goldschmidt

**Affiliations:** 1Division of Paediatric Gastroenterology, Hepatology and Liver Transplantation, Department of Paediatric Kidney, Liver and Metabolic Diseases, Hannover Medical School, 30625 Hannover, Germany; Luisa.Stasch@stud.mh-hannover.de (L.S.); Baumann.U@mh-hannover.de (U.B.); konietzny.christiane@mh-hannover.de (C.K.); pfister.eva-doreen@mh-hannover.de (E.-D.P.); goldschmidt.imeke@mh-hannover.de (I.G.); 2Medical Psychology, Hannover Medical School, 30625 Hannover, Germany; ernst.gundula@mh-hannover.de (G.E.); lange.karin@mh-hannover.de (K.L.); sautmann.kirsten@mh-hannover.de (K.S.)

**Keywords:** paediatric liver transplantation, health-related quality of life, PedsQL, ULQUI, parental education

## Abstract

Objective: Structured education programs have been shown to improve somatic outcome and health-related quality of life (HRQOL) in a variety of chronic childhood diseases. Similar data are scarce in paediatric liver transplantation (pLTx). The purpose of this study was to examine the relationship of parental disease-specific knowledge and psychosocial disease outcome in patients after pLTx. Methods: Parents of 113 children (chronic liver disease n = 25, after pLTx n = 88) completed the transplant module of the HRQOL questionnaire PedsQL, the “Ulm quality of life inventory for parents of children with chronic diseases” ULQUI, and a tailor-made questionnaire to test disease-specific knowledge. Results: Parental knowledge was highest on the topic of “liver transplantation” and lowest in “basic background knowledge” (76% and 56% correct answers respectively). Knowledge performance was only marginally associated with HRQOL scores, with better knowledge being related to worse HRQOL outcomes. In contrast, self-estimation of knowledge performance showed significant positive correlations with both PedsQL and ULQUI results. Conclusion: Patient HRQOL and parental emotional wellbeing after pLTx are associated with positive self-estimation of parental disease-specific knowledge. Objective disease-specific knowledge has little impact on HRQOL. Parental education programs need to overcome language barriers and address self-efficacy in order to improve HRQOL after pLTx.

## 1. Introduction

The focus of care for children after liver transplantation has moved in recent years from survival to reduction of long-term comorbidities [1,2,3]. Infections, chronic graft hepatitis, metabolic syndrome, cardiovascular complications and skin cancer all represent somatic long-term complications of liver transplantation in childhood [4,5,6]. At the same time, health-related quality of life and psychosocial rehabilitation have gained more importance in follow-up care [1,7,8].

Prevention of comorbidities requires adherence with immunosuppressive medication as much as adherence to lifestyle recommendations [9,10]. In other chronic diseases of childhood, such as diabetes or asthma, great success has been achieved by introducing structured education programs for children and their parents [11,12,13,14,15]. For these disease entities, improvement of both somatic outcome and health-related quality of life (HRQOL) through structured education interventions has been repeatedly demonstrated. Only limited data are available on the effects of structured education interventions in paediatric liver transplantation. Lerret et al. have examined whether pre-discharge education and parental perception of readiness for discharge are associated with a reduction in coping difficulties and health-care utilization in children after solid organ transplantation [16]. Readiness for discharge was related to families’ capacity to cope at home, but was not associated with somatic outcomes such as unplanned visits to the emergency department or unplanned hospital visits [16]. In their study, quality of discharge teaching was assessed by parental perception. No formal assessment of parental disease-specific knowledge was made.

The purpose of our study was to examine the relationship of parental disease-specific knowledge and both somatic and psychosocial outcome. To this end, we conducted a cross-sectional study in parents of children awaiting or having undergone liver transplantation. Parental disease-specific knowledge was examined in relation to medication adherence and parental and patient HRQOL. The overarching aim of this study was the identification of relevant knowledge areas that contribute most to patient outcome and merit further attention in parent education interventions.

## 2. Materials and Methods

### 2.1. Subjects

Between September 2019 and February 2020 113 parents/caregivers of 113 patients (68 girls, 45 boys) aged 6 months to 18 years (median 8.7 years) with chronic liver disease (n = 25) or after liver transplantation (pLTX) (n = 88) were recruited from our out-patient clinic. An overview about patient eligibility and enrolment is given in Figure 1. Participants’ demographic data are summarized in Table 1. Inclusion criteria were age 0–18 years plus diagnosis of advanced chronic liver disease or history of liver transplantation in the corresponding children. Exclusion criteria were missing informed consent, parent’s/caregiver’s lack of ability to understand the German language or failure to complete the questionnaires.

### 2.2. Assessment of Disease Specific Knowledge

Disease-specific knowledge was assessed using a tailor-made questionnaire. A non-validated translation of this questionnaire is presented in Appendix A. Disease-specific knowledge was examined by a set of eleven multiple choice questions with up to 4 correct answer possibilities each. These multiple choice questions were supplemented by a list of 23 individual statements that had to be rated as “true” or “false”. For each question, parents also had the option to tick “I don’t know”. Parents’ answers were rated as correct/not correct, and the percentage of correct answers was calculated. “I don’t know” was counted as equivalent to “wrong”. Cronbach’s alpha as a measure of internal consistency and scale reliability was high for both subscales (0.82 and 0.91 respectively).

In addition to actual knowledge, self-assessment of knowledge was examined using a 17-item list of different knowledge topics where parents could rate their own knowledge on a 5-step Likert scale, ranging from 1: very little knowledge to 5: very good knowledge. Both actual knowledge and self-assessment items were grouped into four different topic areas, namely: basic knowledge on liver functions and chronic liver disease (including complications such as portal hypertension), liver transplantation including information on waiting list procedures, medication knowledge, and follow-up care.

### 2.3. Health-Related Quality of Life of Patients and Emotional Wellbeing of Parents

Parental emotional wellbeing and disease specific burden was assessed by the validated Ulm Quality of Life Inventory for Parents of chronically ill children (ULQUI) [17]. Patients’ health related quality of life (HRQoL) was measured using the transplant module of the established health-related pediatric quality of life questionnaire (PedsQL) [18].

The PedsQL^TM^3.0 Transplant Module consists of 8 subscales, namely: (1) Medicines I (9 items; barriers to medical regimen adherence), (2) Medicines II (8 items; medication side effects), (3) Transplant and Others (8 items; social relationships and transplant), (4) Pain and Hurt (3 items; physical discomfort), (5) Worry (7 items; worries related to health status), (6) Treatment Anxiety (4 items; fears regarding medical procedures), (7) External appearance (3 items; impact of transplant on appearance), and (8) Communication (4 items; communication with medical personnel and others regarding transplant issues). The parent proxy-report forms assess parents’ perceptions of their child’s HRQOL. Higher values in the PedsQl indicate that this area presents less of a problem.

The Ulm Quality of Life Inventory for parents of chronically ill children (ULQUI) is a 29 item self-report questionnaire specifically designed to depict parental emotional burden. The ULQI consists of five primary scales, namely physical/daily functioning, satisfaction with family, emotional stability, self-realisation and well-being. All raw scores are linearly transformed to a scale of 0–100, where higher scores indicate higher quality of life (QoL). A non-validated translation of the ULQUI can be found in Appendix A.

### 2.4. The Medication Level Variability Index (MLVI)

MLVI was used as a measure for medication adherence as described by Shemesh et al. [19]. MLVI is calculated as the standard deviation of a set of 4–8 tacrolimus or ciclosporin trough blood levels spread over two years retrospectively from the timepoint of the survey. Target trough levels change throughout the 1st year after transplantation and remain stable from one year post Tx onwards. In order to avoid bias from changing target trough levels, only patients with transplantation >3 years ago at the time of study entry were included for MLVI calculation. Patient notes were checked for conditions that might interfere with drug absorption or metabolism during the observation time. Patients with diarrhea, hepatitis b or e during the observation period, or multiple hospitalization for other reasons were excluded from MLVI analysis. The same goes for patients with concomitant medications known to interact with tacrolimus or ciclosporin metabolism (e.g., fluconazole, anticonvulsive drugs).

### 2.5. Statistical Analysis

Categorical variables are described with numbers and frequencies. Continuous variables are given as median with interquartile range or mean with standard deviation depending on distribution. Missing data were not imputed. Correlation between variables was calculated using Pearson’s r. Comparisons between groups (e.g., rejection/non rejection) were made by student’s *t*-test. Significance was set at a two-tailed level of *p* < 0.05. All analyses were conducted using the statistical software package IBM SPSS Version 26.0 (SPSS Inc., Chicago, IL, USA).

### 2.6. Ethical Considerations

This study was approved by the local Ethics Committee (Statement No. 8474) and was conducted in accordance with the Helsinki Declaration on medical research involving human subjects. Informed consent was obtained from participating adolescents and parents/caregivers.

## 3. Results

### 3.1. Parental Disease-Specific Knowledge

Overall, parents achieved 67.5% correct answers. A complete list of items and corresponding results is given in Table 2. In questions with low correct answer rates, rates for “don’t know” far exceeded rates for the wrong answer. Apparently, parents preferred to admit to ignorance rather than simply guess an answer.

When differentiated into topic areas, parents performed best in “general knowledge about liver transplantation” (76.0% correct answers) and “knowledge on follow-up (74.5%, difference n.s.). Knowledge performance was significantly lower in “Medication knowledge” (66.6%, *p* < 0.01) and “general knowledge about liver and liver disease” (55.6%, *p* < 0.01).

### 3.2. Self-Estimation of Disease-Specific Knowledge

Parents had been asked for a self-assessment of their knowledge in the different topic areas. Self-assessment was made by a 5-step Likert scale, from 1: very little knowledge” to 5 “very good knowledge”. Results are given in Table 3. The single items with highest ratings were “medication of my child” (4.1) and “everyday life with the disease” (4.0). Self-assessment of knowledge showed moderate effect size correlation with actual knowledge test results in basic liver knowledge (r = 0.45), transplantation (r = 0.40) and medication (r = 0.48, *p* < 0.01 respectively), and a weak effect size correlation in the area of follow-up care (r = 0.33, *p* < 0.01).

### 3.3. Is the Quality of Parental Disease-Specific Knowledge Associated with Outcome?

#### 3.3.1. MLVI and Rejection

MLVI was used as surrogate measure for medication adherence. MLVI was 14 ± 15.4 ng/mL in Cyclosporin-A (CSA) users (n = 24) and 1.1 ± 1.1 ng/mL in Tacrolimus users (n = 53). Performance in the disease-specific knowledge tests showed no significant correlation with the medication level variability index (MLVI) neither in CSA nor in Tac users (data not shown). A threshold of MLVI 2.5 for Tac users has been reported as a predictor of acute rejection and as a measure for non-adherence [19]. Knowledge performance did not differ between parents whose children had MLVI above or below this threshold (data not shown). Similarly, a history of rejection or re-transplantation were not associated with differences in these knowledge test results.

#### 3.3.2. Psychosocial Outcome

PedsQL und ULQUI results are given in Table 4 and Table 5.

ULQUI and PedsQL scores showed weak to moderate effect size correlation with each other (r = 0.374, *p* < 0.001), indicaing that in general parental wellbeing was associated with HRQOL of the transplanted child.

For both PedsQL and ULQUI, a number of significant negative correlations was found between psychosocial scores and knowledge test performance (Table 6 and Table 7, upper half respectively). While these correlations show only small to moderate effect sizes, they appear to indicate that better parental knowledge is associated with worse perception of the child’s wellbeing, as well as with reduced parental wellbeing.

In contrast, parental estimation of their own knowledge showed numerous positive correlations of small to moderate effect size with both the PedsQL and the ULQUI (Table 6 and Table 7, respective bottom half). In particular, parents with a higher self-assessment of their own disease-specific and transplantation knowledge ascribed higher scores to their children’s HRQoL regarding medication, treatment anxiety and communication. Estimated knowledge on follow-up care also correlated positively with ULQUI summary score (r = 0.21, *p* = 0.03) as well as ULQUI subscores for physical/daily functioning (r = 0.25, *p* = 0.01), satisfaction with family life (r = 0.23, *p* = 0.01) and general wellbeing (r = 0.20, *p* = 0.04) (Table 7). While these associations only have small effect sizes, the notion that estimated knowledge and HRQOL are positively correlated is supported by the finding of significantly higher ULQUI and PedsQL scores in parents with self-estimated knowledge in the upper tertile, compared to the lower 2 tertiles (Table 8). This difference could only be demonstrated for self-estimated knowledge, but not for knowledge test results (data not shown).

### 3.4. Which Other Factors Affect Knowledge and Psychosocial Outcome?

Knowledge test performance was significantly better in German native speakers. In contrast, self-estimated knowledge did not differ between native speakers and non-native speakers.

Parents with higher educational background achieved higher knowledge test performance in the area of basic knowledge on liver and liver disease only. No differences according to parental education status were found in the other topic areas.

Parents living in a stable relationship scored higher in most ULQUI subscales compared to parents without a partner, but did not achieve higher PedsQL scores. Similarly, single parents had lower ULQUI scores compared with parents with shared parental responsibilities. PedsQL pain scores were lower in single parent families, whereas PedsQL medication side effect scores were higher. Being unemployed was associated with lower ULQUI scores for emotional burden and self-realisation, and with lower PedsQL Medication scores.

## 4. Discussion

The purpose of our study was to examine the relationship of parental disease-specific knowledge with both somatic and psychosocial outcome. We found parental disease-specific knowledge to be satisfactory, particularly in the areas of liver transplantation and follow-up care. Parental knowledge test performance correlated moderately with parental self-estimation of knowledge. Contrary to our expectations, knowledge test performance was negatively correlated with both parental and patient psychosocial wellbeing. Whether increased knowledge fosters heightened awareness for problems, or whether ongoing health-related problems drive the acquisition of more knowledge, remains a matter of speculation. In contrast to actual disease-specific knowledge, parental self-estimation of knowledge was positively correlated with both parental psychosocial wellbeing as well as with parental perception of patients’ HRQOL.

Neither actual nor estimated knowledge correlated with somatic outcome in our study. We chose medication variability index as a surrogate parameter for adherence and medical outcome. MLVI has been well described as a marker for adherence, which in turn is vital for long-term outcome [20,21]. MLVI above the cut-off of 2.5 ng/mL (in Tacrolimus users) was predictive of the occurrence of acute cellular rejection [10,19]. The choice of MLVI as somatic outcome parameter has several potential limitations, including variations in target levels over time and a strong inverse correlation with both age and time elapsed since transplantation. Most important, MLVI only represents a fraction of potential somatic outcomes. However, suitable somatic outcome parameters for this type of study are difficult to find. Unlike diabetes for instance, where HbA1c represents an easy-to-determine and fast-reacting outcome parameter that will reflect changes in disease management and adherence in a timely fashion, an equivalent easy to use biochemical outcome marker does not exist in pLx. Further (interventional) studies on long-term effects of parental education programs should aim to include additional parameters such as growth, physical activity reporting, rejection rates and use of medical ressources.

A few limitations need to be addressed. We used a tailor-made questionnaire to assess parental disease-specific knowledge with multiple-choice questions and statements to be rated as true or false. Any tailor-made solution carries the risk of representing a subjective selection of topics influenced by the study team’s preferences. In addition, wording and structure of the questions might be unintelligible for some parents, depending on language or academic background. To the best of our knowledge, no standardized instrument exists to assess parental disease-specific knowledge in the context of paediatric liver transplantation. Topics of questions were drawn from daily clinical practice and incorporated issues raised during routine clinic visits, during counseling sessions and in patient-led phone calls. Correct answer rates ranged from 46% to 91%. Translated into exam pass grades, this would correspond to marks between D and B. A clear definition of what constitutes an acceptable level on health-related knowledge does not exist. Indeed, such a target level of knowledge would need to be defined in order to evaluate the effects and applicability of a new structured disease education program.

Perhaps the most striking finding of our study is the fact that parental well-being (and perception of patient well-being) did not correlate positively with actual knowledge, but with self-estimated knowledge.

Good self-assessed knowledge on basic liver function, transplantation, medication and follow-up care was associated with better scores in medication subscales and total PedsQL summary score. Similarly, higher estimated knowledge on follow-up care was associated with improved physical functioning, satisfaction with family life, general wellbeing and total summary score of the ULQUI. This strong effect of positive self-assessment is reflected in other studies on parental disease-specific education. In the study by Lerret et al., “readiness for discharge” is estimated using a self-perception inventory, not an objective knowledge-based test [16]. Readiness for discharge correlates strongly with coping at home in this study. These findings illustrate that disease-specific education for parents should target their sense of self-efficacy [22] and internal control at least as much as actual disease-related skills. In the knowledge tests, questions targeting “need to do” topics such as vaccinations, or taking of immunosuppressive education, had higher response rates than questions regarding theorectical background knowledge. This finding again illustrates parental preferences for practical knowledge and for “feeling ready to cope”. Evaluation of education programs in children with variable chronic illnesses show an increase in HRQoL after training particularly if training efforts center on strengthening parents’ and patients’ self-monitoring, problem-solving skills, and thus their sense of self-efficacy [15]. These findings underline the importance of patient empowerment in all educational efforts.

## 5. Conclusions

Success of parental training in paediatric liver transplantation can be difficult to assess when using a knowledge-based approach, since validated instruments are lacking and consent on what constitutes acceptable knowledge levels does not exist. Knowledge test results in our study suggest a preference for practical, “how to do” knowledge. Parental perception of their own knowledge has significant positive associations with both parental and patients’ psychosocial wellbeing. This finding emphasizes the necessity to incorporate parental self-perception, self-management skills and self-efficacy as targets of parental educational interventions.

## Figures and Tables

**Figure 1 children-09-00098-f001:**
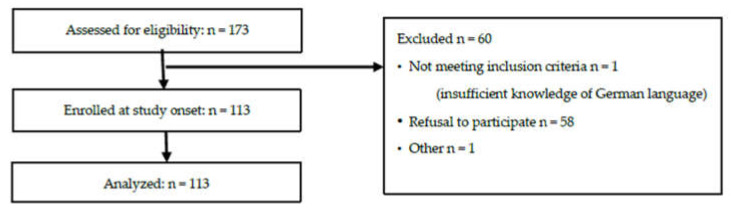
Patient enrolment and attrition.

**Table 1 children-09-00098-t001:** Characteristics of the study sample.

Patients (n = 113)	n (%)	Parents (n = 113)	n (%)
**Sex**		**Primary caregiver’s marital status**	
Male	45 (39.8)	information missing	4 (3.5)
Female	68 (60.2)	Single-parent household	14 (12.4)
**Age at study entry (years)**		Two-parent household	95 (84.1)
Mean SD	9.21 ± 4.87	**Primary caregiver’s highest level of education**	
Median Min, Max	8.67 0.50, 18.25	information missing	4 (3.5)
**Transplantation status**		No degree	2 (1.8)
awaiting transplant	25 (22.1)	General certificate	12 (10.6)
after liver transplantation	88 77.8)	Secondary school	48 (42.5)
**Age at transplant**		Highschool	25 (22.1)
<1 year	38 (43.2)	University education	22 (19.5)
1–4 years	32 (36.4)	**Native language**	
5–12 years	17 (19.3)	German	82 (72.6)
13–17 years	1 (1.1)	other	27 (23.9)
**Time since transplant (years)**		information missing	4 (3.5)
Mean (SD)	7.45 ± 4.53		
Median Min, Max	7.17 0.00, 17.00		
**Primary diagnosis**			
Biliary atresia	70 (61.9)		
PFIC	5 (4.4)		
Acute liver failure	10 (8.8)		
Metabolic	4 (3.5)		
Tumor	6 (5.3)		
Autoimmune Hepatitis	1 (0.9)		
Other	13 (11.5)		

**Table 2 children-09-00098-t002:** Knowledge test performance in multiple choice and single statement questions.

Multiple Choice Questions (4 Possible Answers, Multiple Selection Possible)	Frequency of Correct Answers (%)		I Do Not Know Frequency (%)
Which functions of the liver do you know?	81		3.5
What is the bile’s function?	74		9.2
What are warning signs for biliary tract infection?	77.5		4.5
What is abdominal dropsy?	62.5		18.3
What does the term “liver remodeling” mean?	77.0		16.4
What does cholestasis mean?	66.5		27.0
What is portal hypertension?	48.8		29.1
What can happen in portal hypertension?	38.5		43.1
In case of which diseases in the immediate environment of the child (family, school, day care) should you report to your transplant team?	69.3		18.9
Which possible side effects of immunosuppressive drugs do you know?	58.8		23.4
Which statements concerning the liver transplant waiting list are correct?	77.0		8.1
**Individual Statements to Be Marked as True or False**	**Correct Answer Frequency (%)**	**Wrong Answer Frequency (%)**	**I Do Not Know Frequency (%)**
My child must reach a certain age to be able to get a new liver.	91.0	0.9	8.1
Children who have been on the transplant waiting list the longest get a new liver first.	77.1	10.1	12.8
The children who need a new liver most urgently get a new liver first.	80.7	10.1	9.2
It is possible that a child receives an organ offer although another child has more points on the waiting list.	56.9	11.9	31.2
My child should be as fully vaccinated as possible before transplantation.	77.1	11.0	11.9
Any vaccination may be performed immediately after transplantation.	81.1	4.5	14.4
Inactivated vaccines may be used from one year after transplantation	61.5	4.6	33.9
The majority of organ donations are cadaveric donations from patients who have been diagnosed with brain death.	58.7	4.6	36.7
There is a possibility that a relative donates a part of his/her liver.	96.4	0	3.6
The immunosuppressive drugs are needed to prevent the patient’s own immune system from attacking the “new liver”.	91.7	0.9	7.3
The immunosuppressive drugs must be taken for the whole life.	82.9	1.8	15.3
The immunosuppressants should be taken during meals.	71.4	3.6	25.0
Immunosuppression leads to an increased susceptibility to infections.	73.9	0.9	18.9
The immunosuppressive drug trough level is important because it determines the dose that must be taken.	82.9	0.9	16.2
If the drug trough level is too high, there is a risk of rejection.	45.5	17.3	37.3
My child is not allowed to eat before blood is drawn to determine the level of immunosuppressants.	57.7	21.6	20.7
My child must not take his immunosuppressive drugs before blood is drawn to determine the drug trough level.	76.6	3.6	19.8
In case of “rejection” the immune system reacts to the “new liver”.	81.1	0.9	18.0
Rejection means that the transplanted organ is lost.	64.0	17.1	18.9
Rejection may be related to too much immunosuppression.	41.4	8.1	50.5
It is sufficient if my child goes for a medical checkup once a year.	78.4	7.2	14.4
After transplantation, my child is entitled to a severely disabled person’s card.	80.4	4.6	15.0
My child must never do sports again after transplantation.	91.1	0.9	8.0

**Table 3 children-09-00098-t003:** Self-assessment of disease-specific knowledge by parents.

How Would You Rate Your Level of Knowledge on the Following Topics?	Mean Likert Scale Result
Illness of your child	3.96 ± 0.7
Anatomy of the liver (portal vein, bile ducts, etc.)	3.30 ± 0.8
Functions of the liver	3.42 ± 0.8
Technical terms such as liver remodeling, tarry stools, portal hypertension, etc.	3.17 ± 0.9
Proceedings during an inpatient stay	3.77 ± 1.0
Transplantation	3.51 ± 1.0
Proceedings before liver transplantation	3.37 ± 1.1
Proceedings during the in-patient stay for liver transplantation	3.5 ± 1.1
Proceedings after liver transplantation	3.55 ± 1.1
Meaning of laboratory values	3.42 ± 0.9
Medication of your child	4.09 ± 0.7
Immunosuppressants of your child	3.81 ± 1.2
Diagnostic procedures, e.g., ultrasound, MRI, X-ray	3.75 ± 0.8
Everyday life with the disease	4.04 ± 0.8
Psychological support options	2.95 ± 0.9
Possible assistance for the care of your child (e.g., care allowance, severely handicapped certificate, etc.)	3.10 ± 1.1
Importance of nutrition	3.60 ± 0.9
**Topic Areas**	
Basic knowledge on liver and liver disease (a–d)	3.46 ± 0.7
Transplantation (e–i)	3.54 ± 0.9
Medication (k, l)	3.96 ± 0.8
Follow-up care (j, m–q)	3.49 ± 0.6

Items were rated on a 5-point Likert scale from 1: very little knowledge to 5: very good knowledge.

**Table 4 children-09-00098-t004:** ULQUI summary scores.

ULQUI Subscale	Subscale Summary Score
Physical and daily functioning	68.0
Satisfaction with family life	83.1
Emotional burden	67.3
Self realisation	50.3
General wellbeing	71.4
ULQUI total score	67.4

**Table 5 children-09-00098-t005:** PedsQL transplant module summary scores.

PedsQL Transplant Module Subscales	Subscale Summary Score
About his/her medicines I	86.2
About his/her medicines II (side effects)	87.1
Transplant and Others	74.9
Pain and Hurt	79.1
Worry	80.8
Treatment anxiety	69.7
Preceived physical appearance	79.9
Communication	74.9
PedsQL total score	80.5

**Table 6 children-09-00098-t006:** Correlation of knowledge test results (upper half) and estimated knowledge (bottom half) with PedsQL results.

	Knowledge Test Result	Liver Function and Liver Disease	Transplantation	Medication	Follow-Up Care
**PedsQL transplant module**	About his/her Medicines I	−0.02	0.11	0.03	0.09
	About his/her Medicines II	−0.16	−0.07	**−0.26 ***	−0.2
	Transplant and others	−0.07	0.01	−0.07	−0.04
	Pain and Hurt	−0.17	−0.09	**−0.25 ***	**−0.22 ***
	Worry	−0.01	0.01	−0.06	0.02
	Treatment anxiety	−0.08	−0.01	−0.19	**−0.21 ***
	Perceived physical Appearance	−0.21	−0.01	−0.16	−0.02
	Communication	−0.09	0.03	−0.06	−0.13
	PedsQL Total score	−0.13	−0.01	−0.17	−0.12
	**Self-Assessment of Knowledge**	**Liver Function and Liver Disease**	**Transplantation**	**Medication**	**Follow-Up Care**
**PedsQL transplant module**	About his/her Medicines I	**0.22 ***	**0.39 ****	**0.34 ****	**0.31 ****
	About his/her Medicines II	0.04	0.14	0.15	0.25 *
	Transplant and others	0.13	0.17	0.17	0.20
	Pain and Hurt	0.03	0.09	0.02	0.01
	Worry	0.08	0.20	0.04	0.21
	Treatment anxiety	0.15	**0.24 ***	0.18	0.33
	Perceived phyiscal Appearance	−0.02	0.06	0.07	0.17
	Communication	0.04	0.12	0.09	0.24 *
	PedsQL Total score	0.14	**0.27 ****	**0.22 ***	**0.33 ****

* *p* ≤ 0.05, ** *p* ≤ 0.01, bold was used to highlight the values that are statistically significant.

**Table 7 children-09-00098-t007:** Correlation of knowledge test results (upper half) and estimated knowledge (bottom half) with ULQUI.

	Knowledge Test Result	Liver Function and Liver Disease	Transplantation	Medication	Follow-Up Care
**ULQUI**	Physical/daily functioning	0.06	0.08	−0.03	0.01
	Satisfaction with family life	−0.05	0.01	−0.11	−0.04
	Emotional burden	−0.03	0.10	−0.07	−0.06
	Self realisation	−0.01	**−0.22 ***	−0.17	**−0.21 ***
	General well-being	0.09	0.05	−0.04	−0.05
	ULQI Total score	−0.04	−0.04	−0.15	−0.13
	**Self-Assessment of Knowledge**	**Liver Function and Liver Disease**	**Transplantation**	**Medication**	**Follow-Up Care**
**ULQUI**	Physical/daily functioning	0.14	0.08	0.19	**0.25 ****
	Satisfaction with family life	0.02	0.07	0.15	**0.23 ***
	Emotional burden	0.09	0.02	0.04	0.15
	Self realisation	0.05	−0.02	0.03	0.02
	General well-being	0.12	0.09	0.15	**0.2 ***
	ULQI Total score	0.09	0.04	0.11	**0.21 ***

* *p* ≤ 0.05, ** *p* ≤ 0.01, bold was used to highlight the values that are statistically significant.

**Table 8 children-09-00098-t008:** PedsQL and ULQUI scores in upper and lower tertiles of estimated knowledge scores.

PedsQL	Upper Tertile Estimated Knowledge Score	Lower Two Tertiles Estimated Knowledge Score	*p*
About his/her Medicines I	91.0 ± 10.6	83.0 ± 12.7	<0.01
About his/her Medicines II	90.8 ± 8.7	84.7 ± 15.0	0.02
Transplant and others	79.5 ±14.3	71.7 ± 15.7	0.02
Pain and Hurt	80.3 ± 20.1	78.4 ± 19.3	n.s.*
Worry	85.7 ± 17.1	77.7 ± 20.9	n.s.
Treatment anxiety	79.2 ± 25.5	63.8 ± 29.6	0.01
Perceived phyiscal Appearance	82.4 ± 19.8	77.7 ± 25.2	n.s.
Communication	81.1 ± 23.0	71.0 ± 25.8	n.s.
PedsQL Total score	85.0 ± 10.7	77.4 ± 11.6	<0.01
**ULQUI**			
Physical/daily functioning	72.9 ± 17.1	65.3 ± 17.1	0.03
Satisfaction with family life	89.1 ± 11.4	79.8 ± 17.7	<0.01
Emotional burden	70.1 ± 22.6	65.8 ± 19.5	n.s.
Self realisation	53.6 ± 23.3	48.6 ± 22.4	n.s.
General well-being	75.8 ± 17.9	69.1 ± 17.1	n.s.
ULQI Total score	70.1 ± 16.7	65.4 ± 15.4	n.s.

* n.s. not significant.

## Data Availability

Data are available from the corresponding author on request.

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
