# Peer review of "Parental Disease Specific Knowledge and Its Impact on Health-Related Quality of Life"

_children, 2022, doi:10.3390/children9010098_

Round 1
Reviewer 1 Report
Then Authors present a paper:" Parental disease specific knowledge and its impact on health-related quality of life" that manages an interesting topic already faced as reported by Authors in other chronic diseases. The Introduction as well as Materials and Methods are well reported. Figures and tables are clear and also the Questionnaires in the Appendix. Results should be reported underlining what they want to stress otherwise is a description of the tables and the reader looses himself in capturing the correct meaning. Also Discussion is redundant and the take home message is lacking addressing the results avoiding to give a concrete operative suggestion on how to overcome the problem. It is not necessary to report only the limits of the study but try to give solutions.
Author Response
Thank you for the helpful review!
Following the reviewer's comments, we have shortened and streamlined the results-section in order to avoid redundancies between text and tables, and in order to guide the reader and highlight the most relevant results.
We have also considerably shortened and streamlined the discussion in order to avoid redundancies and focus on what we consider the main finding of our study, which is the fact that parent and patient HRQOL is associated with estimated knowledge, i.e. a sense of self-efficacy, and not necessarily with actual knowledge.
We hope that you will find These changes adeqate in order to improve understanding and readability of our manuscript.
Reviewer 2 Report
In this paediatric liver transplant cohort, parental factual knowledge on basic liver functions, transplantation, medication and follow-up care appears adequate.
This knowledge is not positively associated with somatic outcome parameter or parental and patients’ HRQoL.
In contrast, parental perception of their knowledge has significant positive associations with both parental and patients’ psychosocial wellbeing.
Language barriers have been identified as negative predictors of parental disease specific knowledge.
Future changes towards a more structured educational approach need to address these barriers
Parental self-perception and self-efficacy appear promising targets in order to improve patient and parent HRQOL and self-management skills, and need to be incorporated in educational interventions accordingly

Author Response
Thank you for your kind review. Our understanding of your comments was that you did not require any specific changes to be made. We hope that you will find the adaptations made based on reviewer one's comments acceptable.
Round 2
Reviewer 1 Report
The Authors presented a good revised version of their paper:" Parental disease specific knowledge and its impact on health-related quality of life" improving the quality of the paper. They adequately followed the suggested requests making the paper more comprehensive and clear.